# Controlling the Size and Porosity of Sodalite Nanoparticles from Indonesian Kaolin for Pb^2+^ Removal

**DOI:** 10.3390/ma15082745

**Published:** 2022-04-08

**Authors:** Maria Ulfa, Abu Masykur, Amanah Firdausa Nofitasari, Novia Amalia Sholeha, Suprapto Suprapto, Hasliza Bahruji, Didik Prasetyoko

**Affiliations:** 1Chemistry Education Study Program, Faculty of Teacher Training and Education, Sebelas Maret University, Surakarta 57126, Indonesia; 2Department of Chemistry, Science Faculty, Sebelas Maret University, Jl Ir Sutami 36A, Surakarta 57126, Indonesia; abu.masykur@gmail.com; 3Department of Chemistry, Faculty of Science and Data Analytics, Institut Teknologi Sepuluh Nopember (ITS), Keputih, Sukolilo, Surabaya 60111, Indonesia; amanahfn@gmail.com (A.F.N.); noviaamaliasholeha@gmail.com (N.A.S.); suprapto19@gmail.com (S.S.); 4Centre of Advanced Material and Energy Science, University Brunei Darussalam, Bandar Seri Begawan BE 1410, Brunei; hasliza.bahruji@ubd.edu.bn

**Keywords:** sodalite, kaolin, kinetics, Pb^2+^ adsorption, isotherm

## Abstract

Mesoporous sodalite nanoparticles were directly synthesized from Indonesian kaolin with the addition of CTABr as a mesopore template. The studies highlighted the importance of aging time (3–12 h) and temperature (50–80 °C) on increasing surface area and mesoporosity of sodalite. Indonesian kaolin was used without pre-treatment and transformed to sodalite following the initial molar composition of 10 Na_2_O: 2 SiO_2_: Al_2_O_3_: 128 H_2_O. Characterization data revealed the formation of high surface area sodalite with mesoporosity at increasing aging temperatures and times. The presence of CTABr as templates produced sodalites nanoparticles with smaller aggregates than the non-template sodalite. The sodalite sample obtained at 80 °C of crystallization temperature for 9 h (S80H9) displayed the highest mesopore volume (0.07612 cm^3^/g) and the highest adsorption capacity of Pb^2+^ (212.24 mg/g). Pb^2+^ was suggested to adsorb via ion exchange with the Na^+^ counter cation and physical adsorption.

## 1. Introduction

Lead is one of the main heavy metal pollutants in industrial wastewater, non-biodegradable, hazardous, and has adverse effects on human health and ecosystems [1,2]. Lead causes severe damage to the nervous system, kidney function, and reproductive organs and is severely harmful to cardiovascular, hematopoietic, renal, and gastrointestinal systems [3,4]. Several methods have been investigated for the removal of lead from wastewater, such as reverse osmosis [5], precipitation [6], ion exchange [7], and electrodialysis [8]. However, such processes require high operational costs and additional treatment for disposal. Adsorption with solid adsorbent offers an efficient solution for wastewater treatment [9,10]. Zeolites as adsorbents have superior performance compared to the other solid adsorbents such as hydroxyapatite [11], activated carbon [12], and polymer resins [13]. Zeolite has a uniform pore structure and high surface area to ensure efficient diffusion and adsorption of large pollutants [14,15]. Sodalite is a zeolite with good adsorption ability [16] that can be used as a catalyst or catalyst support [17].

Sodalite, Na_8_(Al_6_Si_6_O_24_), has a cubic structure with two cage unit cells. The first cage consists of four rings, and the second cage consists of six rings. The six-member rings form a set of channels parallel to the cube’s diagonals and intersect at the corners and the centers of the cube unit cells to form large cavities. The cavities in sodalites allow diffusion and adsorption of large molecules, either as gas or liquid, such as CO_2_ and heavy metals. The presence of ions such as Na^+^ in sodalite further enhances heavy metals removal via ion exchange. Such properties show the great potential of sodalite as an adsorbent for heavy metals. Sodalite preparation methods include hydrothermal [18], microwave [19], alkali fusion [20], and the templating route [21]. Sodalite can be synthesized using naturally abundant minerals such as kaolin [22], coal fly ash [23], and rice husk ash [24], not only to enhance the potential of minerals but also to reduce the cost of production. In this research, kaolin was used as a precursor for sodalite based on its high abundance in Indonesia at ~343,164,200 tons [25].

Removing heavy metals using a low-cost technology has attracted interest in designing an adsorbent with a large pore structure and high surface area. This study produced mesoporous sodalite from kaolin minerals while using CTABr as a template. The hexadecyl trimethylammonium bromide (CTABr) template is a low-cost structure-directing agent and a porogen due to electrostatic interactions between the main elements of the kaolin framework (Si, Al) and the head group of CTABr. CTABr is a salt with a bromide anion and a quaternary ammonium cation (CTA^+^) consisting of three methyl groups and one hexadecyl group on the nitrogen atom. CTA^+^ contains a hydrophilic head around a positively charged nitrogen atom and a hydrophobic tail (hexadecyl group), giving rise to its surfactant properties. When used in nanoparticles synthesis, the synthetic procedure involves several steps of the final product depending on the exact conditions of synthesis temperature, concentration, time, and acidity.

Kaolin was used without pretreatments, such as calcination or sulfonation, to reduce additional energy that can contribute to the cost of production. In this work, additional insights into the size and porosity of sodalite using a CTABr template were obtained, highlighting the role of mild conditions in time and temperature during the hydrothermal process. For this, sodalite using CTABr at different times and temperatures in the hydrothermal process were studied and compared with that without CTABr to gain insight into the role of mild hydrothermal conditions in the sodalite pore. To the best of our knowledge, the optimization of synthesis conditions on the properties and the adsorption performance of sodalite have been rarely reported. Therefore, the effects of aging time and temperature were investigated to enhance the surface area and the porosity of sodalite. The resulting sodalites were used as adsorbents for heavy metal (Pb^2+^) removal from wastewater.

## 2. Materials and Methods

### 2.1. Materials

Materials used in this study were kaolin (kaolinite, Al_4_(Si_4_O_10_)(OH)_8_), >99%, obtained from natural resources, Bangka Belitung, Indonesia, with the composition of Al_2_O_3_ 22%, SiO_2_ 57%), hexadecyltrimethylammonium bromide (CTABr, C_19_H_42_BrN, ≥98%, AppliChem GmbH, Darmstadt, Germany), NaOH (sodium hydroxide, pellets, >99.5%, AppliChem GmbH, Darmstadt, Germany), sodium aluminate anhydrous (NaAlO_2_, Al_2_O_3_ 50–56%, Na_2_O 40–45%, Sigma Aldrich, St. Louis, Missouri, United States) used as an alumina source, and demineralized water. A stock solution of Pb^2+^ ion with a concentration at 1000 ppm was prepared by dissolving Pb(NO_3_)_2_ solids (lead (II) nitrate, >99%, Merck, Kenilworth, New Jersey, United States) in distilled water.

### 2.2. Synthesis of Sodalite

Sodalite was synthesized using 3 Na_2_O: 2 SiO_2_: 1 Al_2_O_3_: 128 H_2_O of initial molar composition. The hydrothermal temperatures were varied at 50, 60, 70, and 80 °C and the time at 3, 6, 9, and 12 h. NaOH (11.4 g) was dissolved in deionized water (34.1 mL) and stirred for 15 min, followed by the addition of kaolin (3 g). The mixture was added with sodium aluminate (0.78 g) and stirred at room temperature for 6 h. The first stage of the crystallization process was carried out at 50 °C for 3 h. The mixture was cooled down, and CTABr (2.59 g) was added and stirred for 1 h. The second stage of the crystallization process was conducted at 100 °C for 24 h. After the crystallization process, the solid residue was filtered, washed with deionized water until pH 7–8, and dried at 100 °C overnight. CTABr was removed via calcination at 550 °C with a heating rate of 2 °C/min in an N_2_ stream for 1 h and then continued under air flow for 6 h. The sodalite samples were denoted as SxHy, where x refers to temperature (°C), and y refers to time (h), as tabulated in Table 1. Moreover, sodalite without CTABr was synthesized through the same procedure and denoted as SOD.

### 2.3. Characterization

XRD (X-ray diffraction) analysis was performed using Phillips Expert with Cu-Kα (40 kV, 30mA) radiations at a range of 2θ = 5 to 50°. The crystallinity of the mesoporous sodalite relative to sodalite zeolite without the addition of CTABr as standard was calculated using Equation (1).
(1)$$\mathrm{Crystallinity}\;\mathrm{relative}\;\left(\%\right)=\frac{\mathrm{Sum}\;\mathrm{of}\;\mathrm{sample}\;\mathrm{intensity}}{\mathrm{Sum}\;\mathrm{of}\;\mathrm{standard}\;\mathrm{intensity}}\times 100\%$$

Moreover, crystal particle size could be calculated by the Scherrer equation shown in Equation (2), where d is the size of the crystal particle (nm), λ is the wavelength of 0.154 nm, θ is the Bragg angle, and β is the peak width (FWHM) at the diffraction peak (2 1 1) in 2θ = 24.65°.
(2)$$d=\frac{0.9\;\alpha }{B\cos\theta }$$

The functional group of solids was characterized using FT-IR (Fourier transform infrared Shimadzu Instrument Spectrum One 8400S, Shimadzu Corporation, Kyoto, Japan). The morphology of the synthesized solids was determined by SEM (Scanning Electron Microscopy, ZEISS EVO MA 10, ZEISS Group, Oberkochen, Germany) and EDX (Energy Dispersive X-Ray, BRUKER 129 EV, BRUKER Corporation, Billerica, Massachusetts, USA). Before the solid was analyzed with the SEM-EDX instrument, the sample was coated with Pd or Au for 15 s at a pressure of 6 × 10^−2^ mbar and placed on a carbon tape base. Nitrogen adsorption–desorption isotherms of samples were observed by using a Nova-1200, Quantachrome Corporation instrument, Boynton Beach, Florida, United States. Samples was degassed for 3 h at 300 °C, and then nitrogen gas was flowed at 77 K. The specific surface area (S_BET_) was calculated by the BET (Brunauer–Emmet–Teller) equation, and S_meso_, V_meso_, and D_meso_ using the BJH (Barrett, Joyner, and Halend) method [26,27].

### 2.4. Adsorption of Heavy Metal Pb^2+^

Sodalite as adsorbent of heavy metal Pb^2+^ was determined by using AAS (Atomic Absorption Spectroscopy, AA-6800, SHIMADZU Corporation, Kyoto, Japan). The adsorption of Pb^2+^ was carried out by the mixing of 0.05 g of sodalite with 200 mL (50 and 100 ppm) of Pb(NO_3_)_2_ solution [28], closed with wrap plastic directly, and stirred for 24 h at room temperature at pH 5. After the adsorption process, 10 mL of the mixture was filtered at intervals of 1–60 min to investigate the effect of contact time between sodalite and Pb^2+^ ion. Furthermore, Equations (3) and (4) determined the adsorption capacities and the adsorption percentage.
(3)$$q_{e}=\frac{\left(C_{0}-C_{e}\right)V}{m}$$
(4)$$R\;\left(\%\right)=\frac{C_{0}-C_{e}}{C_{0}}\times 100\%$$
where *C*_0_ and *C_e_* (mg·L^−1^) were the concentration of Pb^2+^ solution initially and after the process of adsorption, respectively, V (liter) was the volume of Pb^2+^ solution, and m (gram) was the adsorbent (sodalite) mass. The adsorption kinetics equation was determined based on the pseudo-first order by Lagergren and Ho Mckay [29,30].

## 3. Results

### 3.1. Characterization Catalyst

XRD analysis determines the phase, crystallinity, and particle size of the synthesized solids. The X-ray diffractograms at 2θ = 5–50° are shown in Figure 1, while the relative crystallinity values and the particle size are tabulated in Table 2. The crystalline phase of kaolin as raw material was observed from the peaks at 2θ of 12.6, 20.43, 24.94, 38.46, and 45.54° [31]. Following synthesis at different conditions, kaolin transformed to sodalite based on the diffraction peaks at 2θ = 14.079, 24.419, 31.640, 34.720, and 42.839°, corresponding to the (110), (211), (310), (222), and (330) crystal planes, respectively, in accordance with JCPDS No. 75-0709 [21,32].

Calculation of the crystallinity of sodalite from XRD data showed that the S60H6 sample had the highest crystallinity at 109%, while the S60H3 sample had the lowest crystallinity at 40% (Table 2). The S80H6 sample had the smallest particle size, around 19 nm, while the S70H6 sample had the largest particle size, estimated at ~37 nm. Based on the XRD results, sodalite crystalline phase, crystallinity, and particle size were influenced by the addition of the CTABr template, temperature, and crystallization time.

FT-IR analysis determined the bonding between atoms and molecular vibrations of the sodalite samples (Figure 2). The adsorption bands at ~536 and ~1115–1008 cm^–1^ of kaolin was associated with the Al–O bond vibration in Al[O(OH)]_6_ and the vibration of Si–O–Si bonds [31]. The absorption peak at ~795 and 697 cm^–1^ indicated the vibration of Al–OH, while the Si–O bond vibrational peaks appeared at 469 and 430 cm^−1^ [33,34]. The summary of FT-IR wave number (cm^–1^) for all the sodalite samples is shown in Table 3. Sodalite without the addition of CTABr (SOD) was used as the standard sample. SOD samples showed the typical FT-IR spectra of sodalite, based on peak appearance at 987 cm^–1^, which was ascribed to the T–O–T asymmetry stretching vibration (T = Si or Al) of sodalite [35]. The peaks at 719 and 657 cm^–1^ were assigned to T–O–T symmetry stretch vibrations (T = Si or Al). Al–O–H shape changes were shown at wave number 698 cm^–1^. O–T–O buckling vibrations (T = Si or Al) were shown at wave number 518 cm^–1^. The peak at 461 cm^–1^ indicated changes in the shape of Si–O, meanwhile, the single 4-ring sodalite (S4R) appeared at 422 cm^–1^–430 cm^–1^ [22,36].

Sodalite produced with the addition of CTABr showed a similar pattern as the SOD with the main T–O–T asymmetry stretching vibration (T = Si or Al) at 990 cm^–1^ [12]. The bands at 720 and 660 cm^–1^ corresponded to T–O–T symmetry stretch vibrations (T = Si or Al). The shift to a higher wavenumber was observed on the deformation Al–O–H band at 700 cm^–1^ when CTABr was in the mixtures. Moreover, the band associated with Al–O deformation at 550 cm^–1^ was only observed on sodalite synthesized with CTABr. Previous studies demonstrated the importance of aging during the synthesis of zeolite, which allowed the mixture of CTABr and tetraethyl orthosilicate (TEOS) to sufficiently hydrolyze into alkoxide before undergoing hydrothermal treatment. The resulting alkoxide-generated silicate species, Si(OSi)_3_) was not formed during the aging of the solution. In SOD produced without CTABr, the silica-alumina skeleton bending vibration O–T–O (T = Si or Al) appeared at 518 cm^–1^. However, the deformation vibration (Al–O) band at 550cm^–1^ disappeared, indicating that CTABr has a significant effect on the formation of Al–O deformation in sodalite structures [37]. The band at ~460 cm^–1^ indicated a change in the shape of Si–O. A single ring of sodalite (S4R) was shown at ~ 430 cm^–1^, which implied the presence of single 4-ring sodalite (S4R). This could be seen in the uptake bands that appeared, which are presented in Table 3.

Nitrogen adsorption determines the surface area and the pore size distribution of sodalite samples. The specific surface area was determined using the BET (S_BET_) method, while the pore size distribution was determined using the BJH method. The nitrogen adsorption–desorption isotherms in Figure 3 and Figure 4 showed similar isotherms for all the synthesized solids. Nitrogen molecular adsorption occurred in low quantities at the relative pressures of P/P_0_ = 0 to 0.3. The surface was covered with nitrogen molecules within this range to form a single monolayer absorption. At a relative pressure, P/P_0_ = 0.3 to 0.45, a considerable increase in N_2_ uptake indicated the occurrence of mesoporous filling. The presence of pores on the surface of a solid limited the number of layers of adsorbate and capillary condensation, which caused hysteresis at P/P_0_ = 0.6–1.0. Hysteresis occurs due to the differences in the number of desorbed nitrogen molecules and adsorbed nitrogen molecules at similar P/P_0_. Thus, hysteresis was not a result of meso-sized pores in the solids; instead from was space between the sodalites nanoparticles.

In general, based on the isotherm patterns, all sodalite samples showed type IV isotherm profiles, which are characteristic of mesoporous materials. Mesopores are formed from the rearrangement of skeletal structures during the condensation of tetrahedral silica. Figure 4 shows that samples obtained at 80 °C (S80H6 and S80H9) have a high nitrogen uptake at P/P_0_ = 0.3 compared to the other samples, which implied a higher density of mesopores. The calculated surface area and the pore analyzes are summarized in Table 4. Sodalite produced with the addition of CTABr had a lower surface area and mesoporous volume compared to sodalites without the addition of CTABr (SOD). However, S80H9 had the largest mesoporous surface area and volume among sodalite samples with the addition of other CTABr. Mesoporous volume was relatively decreased on large sodalite particles. These results are consistent with previous research, which indicated that large particles decreased the mesoporous volume, whereas a decrease in particle size resulted in increased mesoporous volume and average smaller pore diameter.

Figure 5 summarizes the effects of changing the crystallization time and temperature on the surface area, pore diameter, and crystallinity of sodalite. Increasing the crystallization time to 9 h increased the surface area and the crystallinity by 65% and 35%, respectively. As the temperature increased from 60 to 80 °C, the pore diameter was reduced by 24%. Moreover, at a temperature of 80 °C, the surface area and the crystallinity increased by 14% and 13%, respectively, at 3–9 h. At higher temperatures, smaller crystallite was formed, which enhanced the interaction with CTABr micelles in the second crystallization stage. Efficient interaction between silicate particles and CTABr is necessary to form mesopores with a high surface area [38].

SEM analysis was used to determine the surface morphology of sodalite, whereas energy dispersive X-ray (EDX) analysis was used to determine the composition of elements. Sodalite samples that were characterized using SEM were SOD and S80H9. In general, the morphology of the two samples was found to be almost similar, consisting of non-uniform aggregates. However, sodalite produced in the absence of CTABr (Figure 6a) had larger particles than sodalite synthesis with CTABr (Figure 6b). In S80H9 samples, the interaction between the aluminosilicate with the hydrophilic tail of CTABr prevented the excess growth of sodalite crystals during the hydrothermal reaction, thus producing a smaller size.

SEM instruments equipped with EDX can be used to determine the composition of elements contained in samples that are observed using SEM. EDX measures the chemical composition on micro and nano scales, where each element will have a specific peak. The elemental mapping of SOD and S80H9 samples are shown in Figure 6, and the element composition is presented in Table 5. The SOD and S80H9 samples consisted of elements used as precursors, namely Si and Al. This showed that no other elements were formed during the synthesis process, indicating no impurities in the synthesis sample. However, the Al and Si compositions of the SOD and S80H9 samples were significantly different, indicating that the addition of the template affected the Si/Al ratio of sodalite.

### 3.2. Adsorption and Kinetic Study of Pb^2+^

In this study, the activity of sodalite for removal of Pb^2+^ was conducted on SOD and S80H9. S80H9 was selected based on the largest surface area, and the mesoporous volume among the other sodalite samples produced using CTABr. The concentration of Pb^2+^ solution was analyzed using AAS, and the amount of Pb^2+^ adsorbed on sodalite is tabulated in Table 6. Figure 7 shows the Pb^2+^ adsorbed from SOD and S80H9 samples at different contact times. Pb^2+^ adsorbed on SOD increased with a longer contact time between adsorbent and adsorbate. Sodalite produced using CTABr showed a faster adsorption rate within the first 5 min than sodalite without CTABr. The result indicates that the presence of large mesopores accelerates Pb^2+^ adsorption. However, there are no significant differences in the adsorption rates between the sodalite samples at much longer contact times. The sodalites reached a steady state at 40 min when using a higher concentration of Pb^2+^ but reached steady states within 1 min of adsorption using low Pb^2+^ concentrations. The observation suggests the ability of sodalite to achieve rapid removal of Pb^2+^ in wastewater. Sodalite produced using CTABr displayed a slightly higher adsorption capacity at 212.24 mg/g than sodalite without CTABr at 202.19 mg/g due to the presence of mesopores and the high quantities of oxygen-containing functional groups.

Figure 8 shows the adsorption of Pb^2+^ on different sodalite samples obtained from variations of crystallization conditions. The rate of adsorption varied depending on the textural properties of sodalite. Variation of crystallization conditions significantly altered the surface area and the porosity of sodalite. Increasing the temperature and the aging time in the six selected samples resulted in an increased mesoporous area to surface area ratio between 15 and 24% (Table 4). Although the initial molar composition of sodalite was similar, changing the temperature and the time for crystallization enhanced the adsorption capacity of Pb^2+^. The capacity of sodalite adsorption in this study was greater than that in our previously reported studies, at 97.15 mg/g [16]. The formation of stable silicate particles was crucial to ensure efficient interaction with CTABr for the formation of larger mesopore cavities. High temperatures and longer crystallization in the first stage of hydrothermal treatment ensured the formation of stable silicate particles. Different sizes of silicalite structures also affected the rearrangement of mesopores following the addition of CTABr [39]. Smaller micelles were formed in the presence of CTABr, resulting in the formation of microparticles with a high surface area. Large mesopore diameter accelerated the diffusion of Pb^2+^ for rapid adsorption. The increased surface area and mesopore volume improved the adsorption capacity from 160.8 to 212 mg/g, resulting in a 24% adsorption improvement.

Adsorption of metal cation on sodalite occurs on the surface between the positive charge metal cation and the opposing charge zeolite surface [40]. Pb^2+^ can occupy the sodalite surface via interaction with Si-O- by displacing the hydrogen in Si-OH (Equations (5)–(7)). Apart from that, Pb^2+^ can also adsorb on sodalite via the cation-exchange process. The cation exchange occurred between Na^+^ ions in the sodalite framework and Pb^2+^ ions (Equation (8)). The mechanism for the exchange of Na^+^ ions with Pb^2+^ ions is shown through the following equations:nSi-OH + Pb^2+^ ↔ (Si-O)n-Pb + nH^+^
(5)
SiO^−^ + PbOH^+^ → SiOPbOH(6)
nSiO^−^ + Pb^2+^ → (Si-O)_n_-Pb(7)
nPbOH^+^ + Na^+^(sodalite) → Pb(sodalite) + Na^+^ + nOH^−^(8)

Table 7 summarizes the kinetic analysis data based on the pseudo-first order and the pseudo-second order equations. The plot between ln(qe − qt) versus t for the pseudo-first order model (Figure 9a) and t/qt versus t plot for the pseudo-second order model (Figure 9b) were carried out to obtain the regression coefficient value related to linearity (R^2^). The pseudo-first order plot showed that the R^2^ values for all the samples were within 0.681–0.836. Meanwhile, for the pseudo-second order plot, the R^2^ value of each sample was within 0.963–0.997, thus indicating that the adsorption of Pb^2+^ on sodalites followed the pseudo-second order kinetic model.

The adsorption of Pb^2+^ ions can also cause hydrolysis of sodalite, which releases silicate ions into the solution. According to the effective ionic radii [41], the radii of the metal ions Na^+^ = 0.99 nm and Pb^2+^ = 0.98 nm. The radii of the guest cations influenced the diffusion within the interconnected intrinsic nanopores/cages of sodalite. Only cations with similar ionic sizes will achieve high ion exchange capacity, as observed on Na^+^ and Pb^2+^. The similarity of Na^+^ and Pb^2+^ sizes reduced the possibility of diffusion of other cations such as Al or Si from the main sodalite framework. The approximately similar ion radii of Na^+^ and Pb^2+^ further prevent the disintegration of the sodalite framework during the diffusion. Adsorption using the cation exchange process further benefits from large mesopore diameter, which allow access into sodalite cavities for adsorption via physical interaction.

This work provides a pathway for transforming natural kaolin into sodalite materials. It also paves the way for a sustainable approach to designing advanced zeolite adsorbents using hydrothermal conditions suitable for low-cost removal of heavy metal pollution. However, analysis of the sodalite following adsorption with Pb^2+^ is important to understand the stability of adsorbent. The results of SEM-EDX S80H9 analysis (Figure 10) after adsorption showed a Pb content of 11% and a C content of 1% *w*/*w*. The presence of carbon cerrusite (PbCO_3_) or hydrocerrusite (Pb_3_(CO_3_)_2_(OH)_2_) formation from the reaction between Pb and CO_2_. The ratio of Pb:C of cerrusite and hydrocerrusite should be within the range of 17.5–25.9. However, the EDX results showed the Pb:C ratio of 11, which suggested Pb adsorption on S80H9 only produced a small amount of cerrusite but without hydrocerrusite formation. Since high amount of Pb was absorbed on sodalite, the SEM analysis revealed the formation of larger crystals of sodalite after Pb^2+^ adsorption than before the adsorption.

XRD analysis of the adsorbed S80H9 sodalite in Figure 11 showed a lower peak intensity compared to the fresh catalysts due to partial saturation of the sample surfaces with Pb^2+^ ions. The formation of cerussite and hydrocerrusite (JCPDS 5-417 and JCPDS 13-0131 [42,43]) were not visible from the XRD analysis that may be due to the low concentration of carbonate ion as determined from EDX analysis. The dissolution of cerussite in the equilibrium point as shown in equation 9 may also contribute to a low concentration of cerussite [44].
PbCO^3^ +2H^+^= Pb^2+^ +2H_2_O+ CO_3_^−^(9)

FTIR analysis (Figure 12) of sodalite following the adsorption of Pb^2+^ ion revealed the formation of cerussite based on the appearance of lead ion peak at 1110 cm^−1^ [42], carbonate ion at 860 cm^−1^ [45], and free OH at 3361 cm^−1^, respectively. However, since the adsorption peak of carbonate ion was low, it can be suggested that a small amount of cerussite was formed on the surface sample that may be played a small role in increasing the adsorption process itself, and this is in line with the EDX and XRD of sodalite after adsorption.

## 4. Conclusions

Synthesis of sodalite nanoparticles from raw Indonesian kaolin was investigated in detail and revealed that the formation of mesopores depends on the crystallization time and temperature. CTABr as a mesopore template produced sodalite nanoparticles with smaller aggregates at 80 °C of crystallization temperature for 9 h. The synthesized sodalite displayed the highest mesopore volume and adsorption capacity of Pb^2+^ (212.24 mg/g) that occur via ion-exchange with Na^+^ counter cation, and physical adsorption. The large mesopore diameter prevented structural disintegration of sodalite by providing efficient Na^+^ and Pb^2+^ diffusion.

## Figures and Tables

**Figure 1 materials-15-02745-f001:**
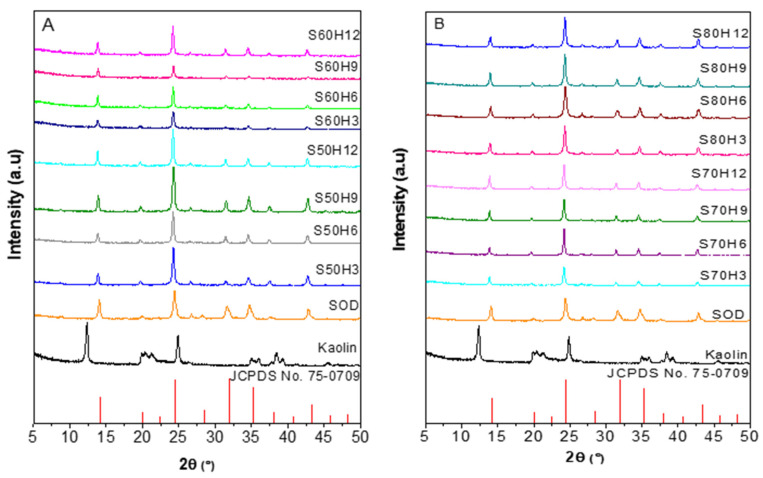
XRD (X-ray diffraction) analysis of samples with various crystallization times and temperatures of (**A**) 50–60 °C, and (**B**) 70–80 °C.

**Figure 2 materials-15-02745-f002:**
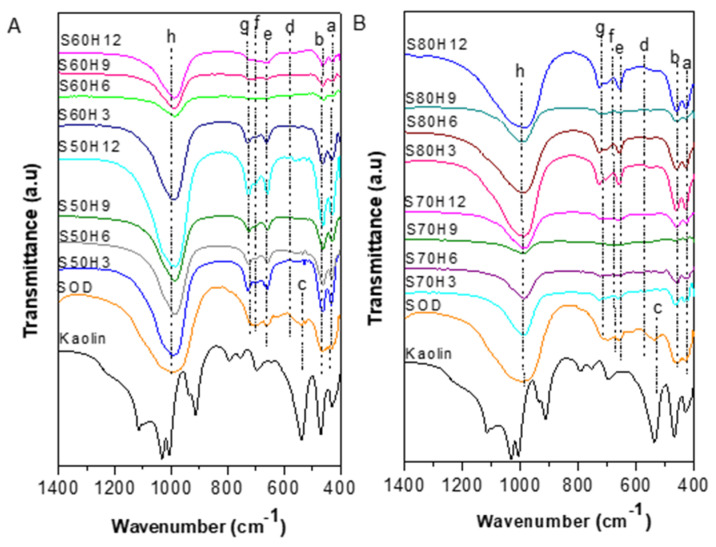
FT-IR (Fourier transform infrared) analysis of samples with various crystallization times and temperatures of (**A**) 50–60 °C and (**B**) 70–80 °C.

**Figure 3 materials-15-02745-f003:**
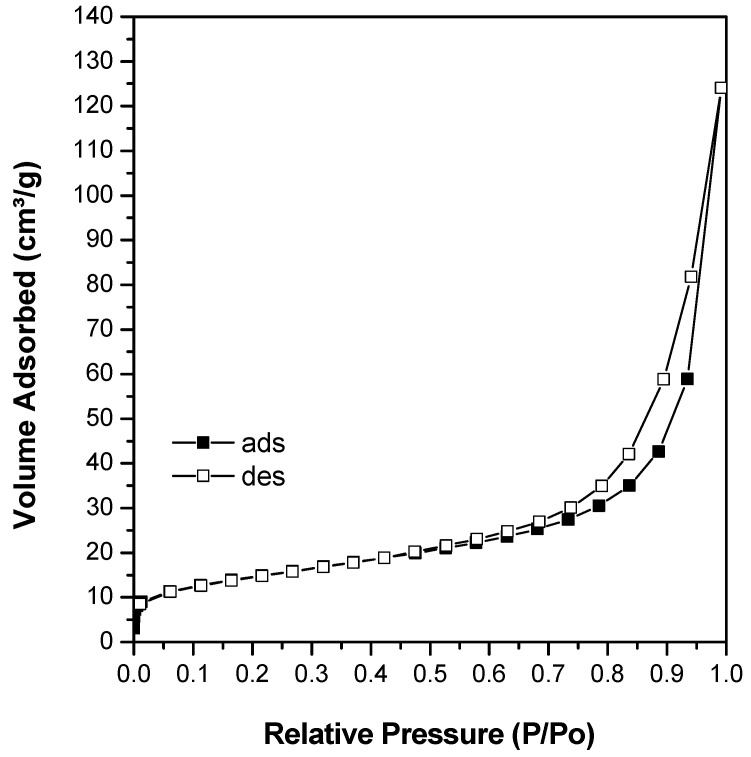
Isotherm of N_2_ adsorption-desorption of sodalite without hexadecyltrimethylammonium bromide (CTABr) addition.

**Figure 4 materials-15-02745-f004:**
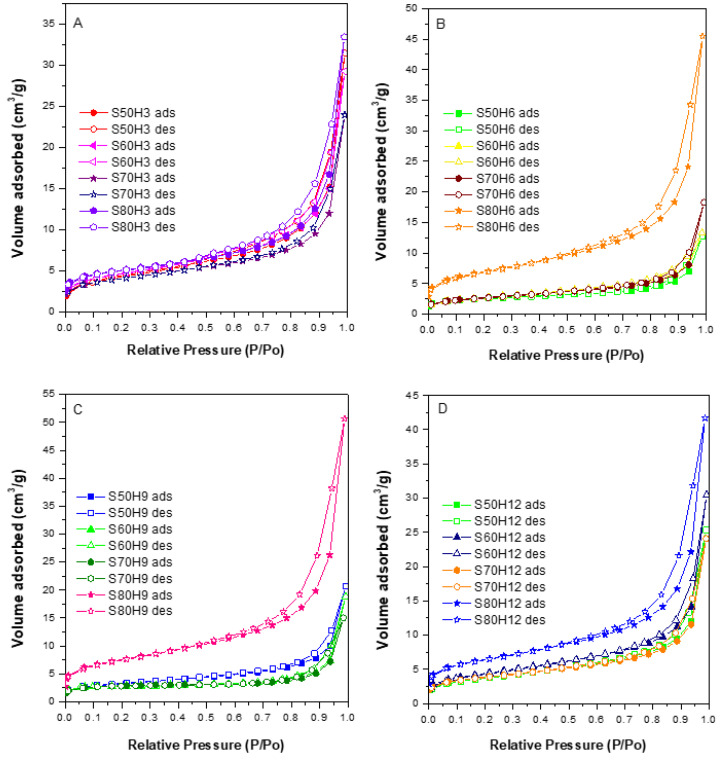
Isotherm of N_2_ adsorption-desorption of sodalite sample with the addition of CTABr with time variation of (**A**) 3, (**B**) 6, (**C**) 9, and (**D**) 12 h.

**Figure 5 materials-15-02745-f005:**
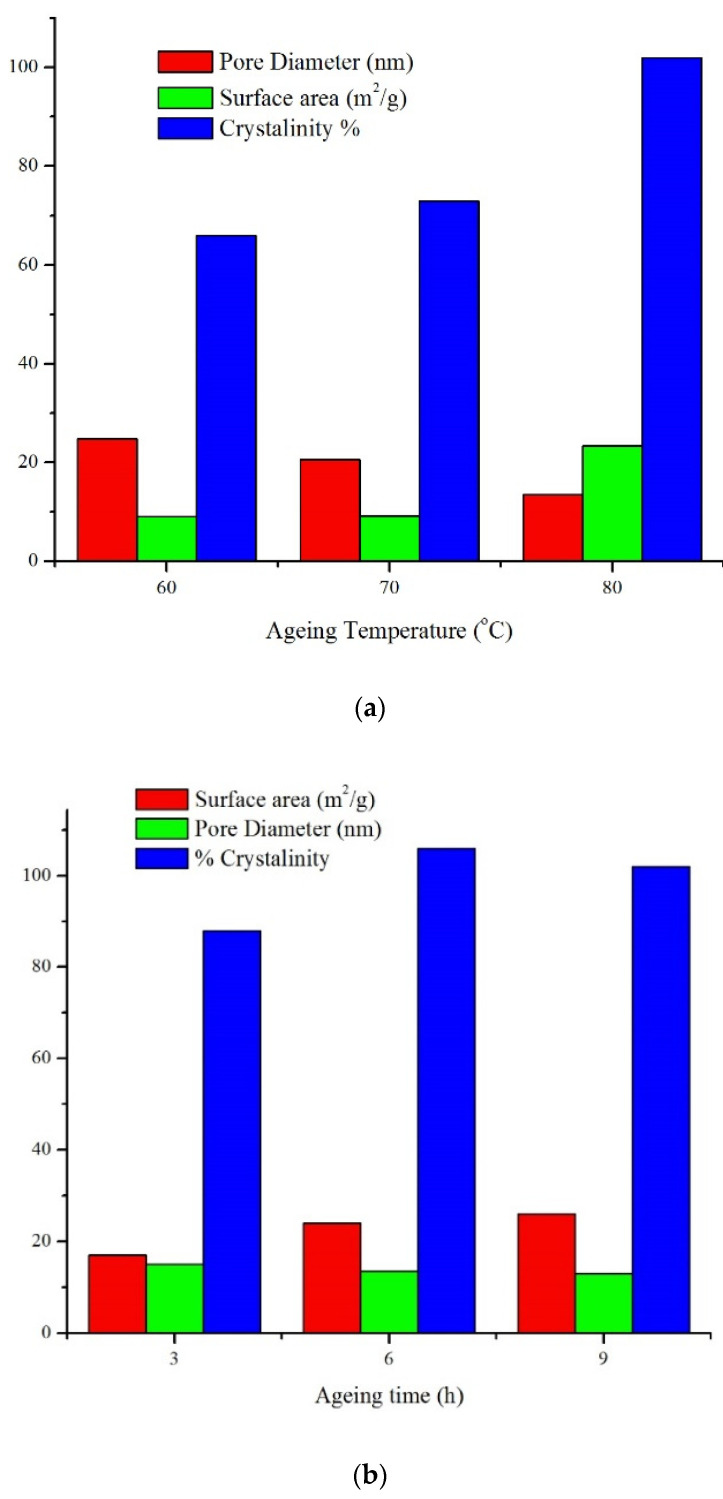
(**a**,**b**) Pore diameter and surface area of sodalite obtained at different temperatures but similar crystallization time—9 h.

**Figure 6 materials-15-02745-f006:**
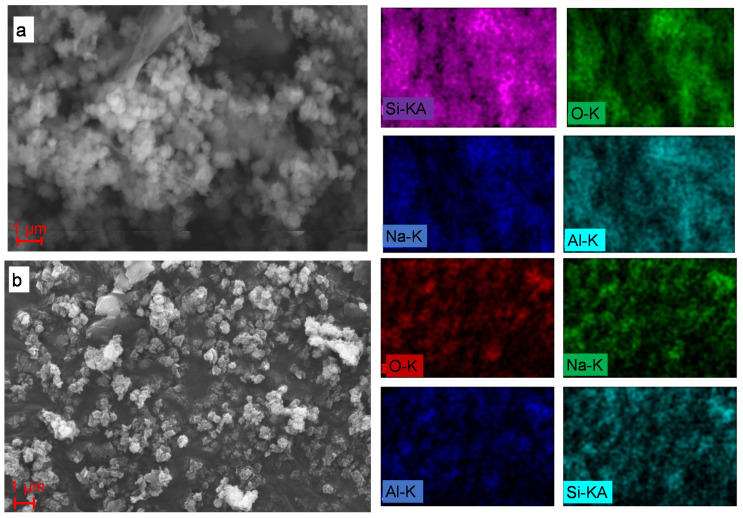
SEM analysis and elemental mapping on (**a**) SOD and (**b**) S80H9.

**Figure 7 materials-15-02745-f007:**
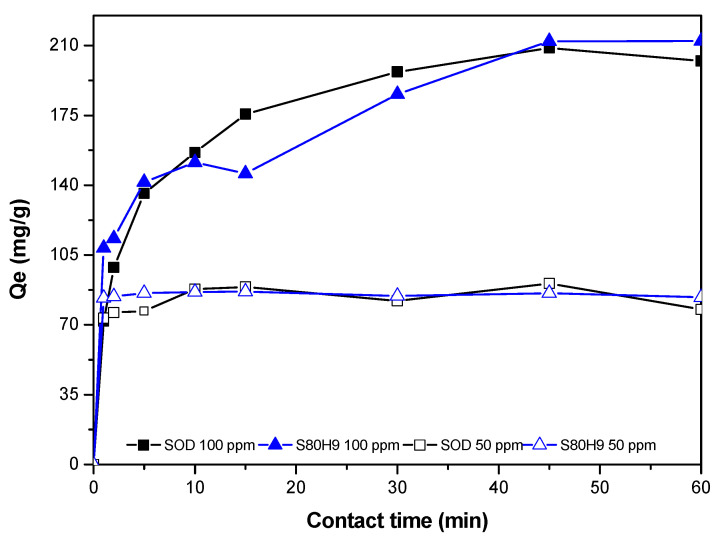
Pb^2+^ adsorption on SOD and S80H9 samples.

**Figure 8 materials-15-02745-f008:**
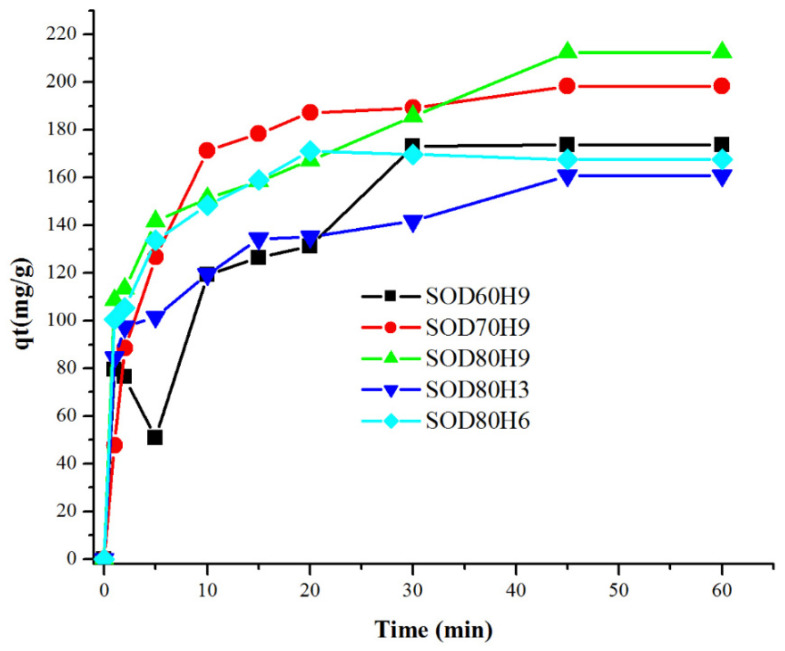
Pb^2+^ adsorption on SOD samples with CTABr (Co = 100 ppm, V = 0.2 L, W = 0.05 g).

**Figure 9 materials-15-02745-f009:**
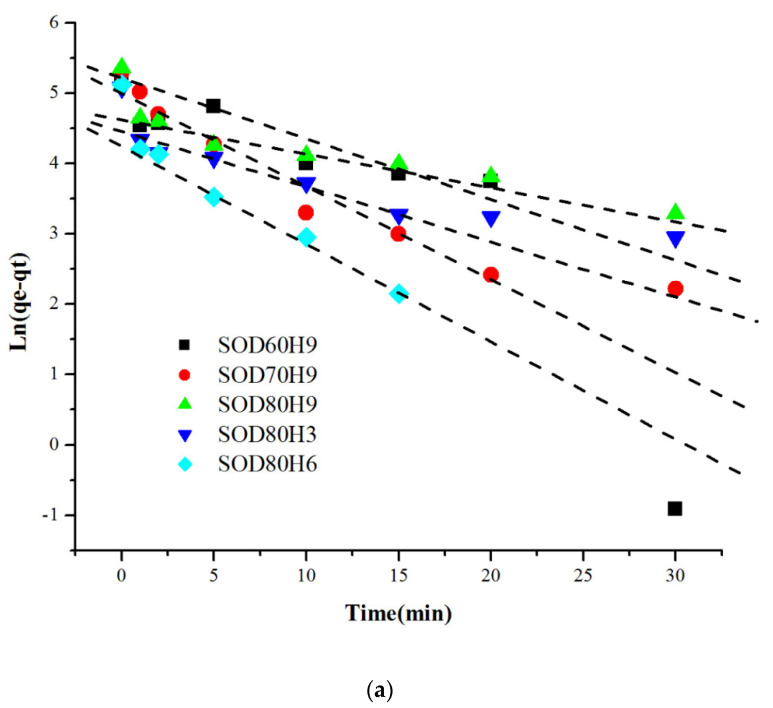
Kinetics of Pb^2+^ adsorption on SOD samples with CTABr with (**a**) pseudo-first-order and (**b**) pseudo-second-order models (Co = 100 ppm, V = 0.2 L, W = 0.05 g).

**Figure 10 materials-15-02745-f010:**
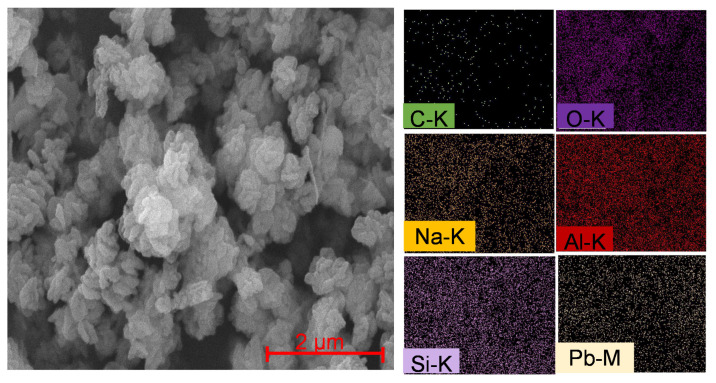
SEM-EDX analysis of S80H9 after Pb^2+^ adsorption.

**Figure 11 materials-15-02745-f011:**
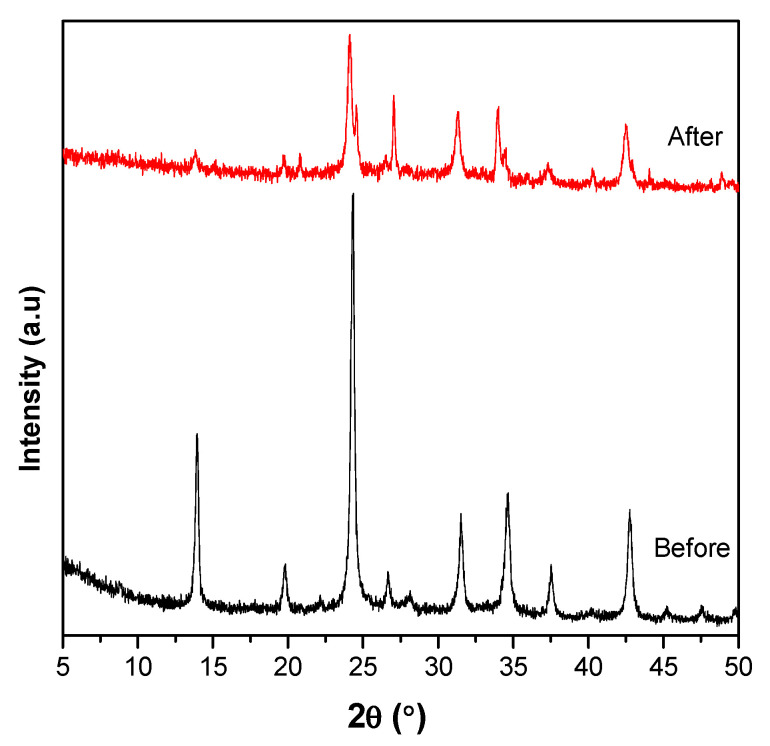
XRD analysis of S80H9 after Pb^2+^ adsorption.

**Figure 12 materials-15-02745-f012:**
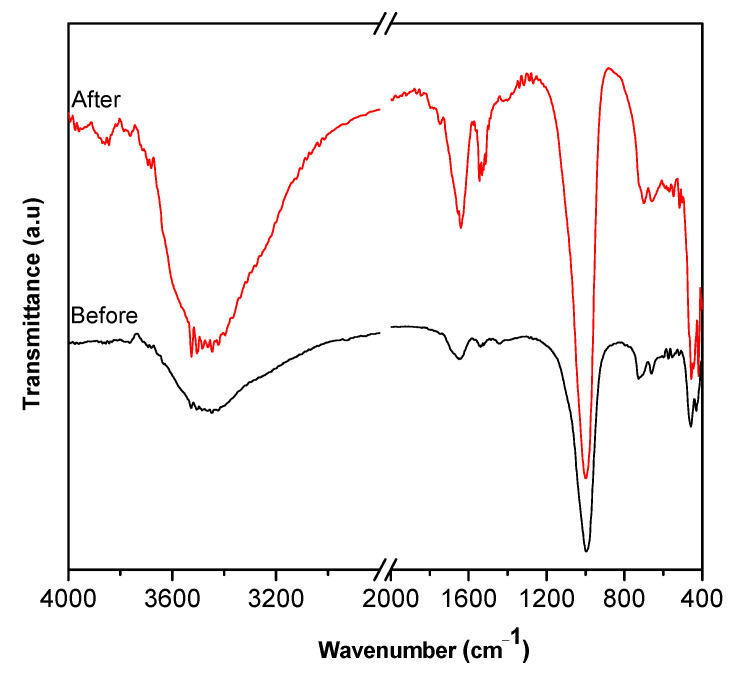
FTIR analysis of S80H9 before and after Pb^2+^ adsorption.

**Table 1 materials-15-02745-t001:** Notation samples in this study with the variation of first crystallization time and temperature.

Temperature (°C)	Time (h)	Code	Temperature (°C)	Time (h)	Code
50	3	S50H3	70	3	S70H3
	6	S50H6		6	S70H6
	9 12	S50H9 S50H12		9 12	S70H9 S70H12
60	3	S60H3	80	3	S80H3
	6	S60H6		6	S80H6
	9	S60H9		9	S80H9
	12	S60H12		12	S80H12

**Table 2 materials-15-02745-t002:** Crystallinity of samples based on the XRD data.

Sample	Intensity at 2θ (°)					Sum	RC (%) ^a^	d (nm) ^b^
	14.00	24.43	31.79	34.86	43.01			
SOD	746	1212	511	564	446	3479	100	17.37
S50H3	390	1274	180	278	384	2506	72	24.12
S50H6	461	1501	331	393	413	3099	89	29.67
S50H9	377	1376	225	361	364	2703	78	29.16
S50H12	318	1636	369	476	479	3278	94	30.95
S60H3	222	608	230	113	214	1387	40	20.08
S60H6	518	1849	429	528	455	3779	109	31.13
S60H9	548	1110	218	239	177	2292	66	26.87
S60H12	414	1424	342	434	433	3047	88	26.3
S70H3	415	969	145	208	199	1936	56	29.27
S70H6	407	1408	285	338	316	2754	79	37.65
S70H9	516	1161	299	301	255	2532	73	32.83
S70H12	676	1288	307	355	298	2924	84	31.47
S80H3	531	1509	206	423	407	3076	88	25.99
S80H6	599	1692	369	544	479	3683	106	19.03
S80H9	676	1629	378	444	419	3546	102	25.78
S80H12	458	1552	366	523	458	3357	96	24.95

^a^ Crystallinity is calculated from the total peak intensity of the sample compared to the number of standard peak intensities (SOD); ^b^ Particle size calculation uses the Scherrer equation.

**Table 3 materials-15-02745-t003:** Wave number IR spectrum of synthesized solid.

Samples	Code of Wavenumber in IR Spectrum (cm^−1^)							
	a	b	c	d	e	f	g	h
SOD	422	461	518		657	698	719	987
S50H3	432	462		555	661	702	729	993
S50H6	430	461		549	659	704	725	987
S50H9	432	461			659	704	725	985
S50H12	432	461		563	659	702	727	985
S60H3	432	462		551	661	704	729	991
S60H6	432	459			663	704	723	987
S60H9	435	461		555	661		727	989
S60H12	432	462		559	657		723	989
S70H3	435	461			663	707	727	991
S70H6	432	459		549	663	702	725	987
S70H9	428	462		570	673		719	993
S70H12	430	461		559	661		725	987
S80H3	430	462		555	661	705	729	989
S80H6	430	461		555	659	705	729	991
S80H9	428	459		555	659	705	729	987
S80H12	430	462			659	705	729	985

a: single 4-ring 4 sodalite (S4R); b: deformation (Si–O); c: bending vibration O–T–O (T = Si or Al); d: deformation (Al–O); e: T–O–T symmetry stretch vibration (T = Si or Al); f: deformation (Al–O–H); g: T–O–T symmetry stretch vibration (T = Si or Al); h: T–O–T asymmetric stretch vibration (T = Si or Al).

**Table 4 materials-15-02745-t004:** Textural properties of samples.

Sample	S_BET_ ^a^ (m^2^/g)	S_meso_ ^b^ (m^2^/g)	S_micro_ ^c^ (m^2^/g)	S_ext_ (m^2^/g)	V_meso_ ^b^ (cm^3^/g)	V_micro_ ^c^ (cm^3^/g)	D_meso_ ^b^ (nm)
SOD	51.87	48.26	4.40	47.47	0.1889	0.0018	15.66
S50H3	15.57	13.40	1.53	14.04	0.0469	0.0006	13.99
S50H6	8.68	5.42	2.46	6.21	0.0182	0.0012	13.44
S50H9	11.27	8.71	2.32	8.95	0.0307	0.0011	14.08
S50H12	13.19	11.55	1.51	11.68	0.0382	0.0007	13.22
S60H3	16.11	13.75	2.77	13.35	0.0440	0.0013	12.78
S60H6	9.54	8.22	0.88	8.66	0.0197	0.0004	9.59
S60H9	9.05	4.29	6.67	2.38	0.0266	0.0034	24.76
S60H12	15.72	12.72	1.15	14.57	0.0454	0.0005	14.26
S70H3	14.00	10.86	2.64	11.37	0.0355	0.0013	13.06
S70H6	9.31	7.24	1.42	7.89	0.0271	0.0007	14.96
S70H9	9.22	3.98	5.36	3.86	0.0206	0.0027	20.66
S70H12	13.40	10.51	2.78	10.62	0.0357	0.0013	13.59
S80H3	17.01	12.64	5.98	11.03	0.0493	0.0030	15.59
S80H6	24.30	20.60	3.49	24.30	0.0680	0.0016	13.21
S80H9	26.39	22.63	4.47	21.92	0.0761	0.0021	13.45
S80H12	22.36	18.32	4.41	17.96	0.0623	0.0021	13.60

^a^ S_BET_, from BET surface area; ^b^ S_meso_, V_meso_, and D_meso_, from BJH method; ^c^ S_micro_, S_ext_, and V_micro_, from t-plot; ^d^ from Scherrer equation.

**Table 5 materials-15-02745-t005:** Elemental composition of sample.

Sample	Element (%)W				Si/Al Ratio
	O	Na	Al	Si	
SOD	61.33	8.44	3.20	2.40	0.75
S80H9	97.50	1.34	0.72	0.43	0.60

**Table 6 materials-15-02745-t006:** Adsorption capacity of Pb^2+^ for SOD and S80H9.

Time (min)	Adsorption Capacity (mg/g) of 100 ppm Pb^2+^		Adsorption Capacity (mg/g) of 50 ppm Pb^2+^	
	SOD	S80H9	SOD	S80H9
1	71.82	108.50	73.26	83.39
2	98.78	113.40	76.16	84.22
5	135.86	141.60	76.94	85.96
10	156.42	151.40	87.84	86.46
15	175.58	145.80	89.05	86.75
30	196.74	185.68	82.05	84.44
45	208.81	212.09	90.79	85.96
60	202.19	212.24	77.74	83.87

**Table 7 materials-15-02745-t007:** Porosity and kinetic results of Pb^2+^ adsorption on SOD samples with CTABr.

	Porosity				First Order			Second Order		
Sample	^a^ S_t_ (m^2^/g)	^b^ S_me_ (m^2^/g)	S_me_/S_t_	q_exp_ (mg/g)	R	K1	Q_cal_	R	k2	Q_cal_
S60H9	9.05	4.29	0.47	160.8	0.790	0.048	7.76	0.995	0.006	140.19
S70H9	9.22	3.98	0.432	167.6	0.821	0.059	8.78	0.994	0.006	140.41
S80H3	17.01	12.64	0.743	173.6	0.782	0.100	4.86	0.963	0.005	145.44
S80H6	24.30	20.60	0.848	198.4	0.681	5.500	5.52	0.997	0.004	155.67
S80H9	26.39	22.63	0.856	212.2	0.836	0.050	8.91	0.989	0.004	175.51

^a^ S_t_, from BET surface area total; ^b^ S_me_, from the BJH method.

## Data Availability

Not applicable.

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
