# Peer review of "Controlling the Size and Porosity of Sodalite Nanoparticles from Indonesian Kaolin for Pb2+ Removal"

_materials, 2022, doi:10.3390/ma15082745_

Round 1
Reviewer 1 Report
Greetings, Editor thank you for providing me with the opportunity to review the article. I reviewed the article with the title ``Controlling the size and porosity of sodalite nanoparticles via purification of Indonesian kaolin as efficient adsorbent for heavy metal Pb2+ removal ``. The present article shows very casually and general information. I am sorry to say, in present form this article should be rejected. Materials is a very highly reputed journal. Although it`s paid journal but still Materials maintain its level.
- Abstract is not giving proper summery of work.
- Introduction is completely massive. No problem statement, no research gap and even novel is in question.
- Results are not well described. Figures are low quality. I am not sure its repeatable results.
- Conclusion section is missing some perspective related to the future research work, quantify main research findings, highlight relevance of the work with respect to the field aspect.
- To avoid grammar and linguistic mistakes, Major level English language should be thoroughly checked. Please revise your paper accordingly since several language issue occurs on several spots in the paper.
However, for author I mentioned some serious flaws here to prepare again the manuscript and submitted to Q2 rank journal. This data is not sufficient for Materials.
Author Response
"Please see the attachment."

Reviewer 2 Report
see attachment

Author Response
"Please see the attachment."

Reviewer 3 Report
This work synthesizes sodalite from kaolinite at various temperatures and crystallization times and tests the removal efficiency of Pb2+ by sodalite from an aqueous solution. The synthesized solids were characterized by a variety of techniques including XRD, FTIR, SEM, EDX, BET, and BJH measurements. The adsorption of Pb2+ was measured by adsorption kinetics rather than adsorption isotherm. The figures and tables look beautiful, but the discussion is not targeted at what has happened behind them but focused on what the figures and tables superficially show. What happened in the synthesis and adsorption means more than what the authors observed by these techniques. In addition, the adsorption of Pb2+ by the synthesized solid, i.e., S80H9, is so close to that by SOD, so it is pointless to compare which is more efficient in terms of Pb2+ removal. Combining with the concerns below, I would not recommend publishing this manuscript in Materials.
General Comments
1) The authors may consider asking for help from English native speakers to improve the clarity of the manuscript. There are too many errors in grammar, tense and number of verbs, punctuations, etc.
2) The justification of this work in the Introduction is weak. The authors may need to introduce more about sodalite, e.g., its structure, properties, and applications (i.e., why was sodalite chosen?).
3) It is not clear how the experiments were conducted and/or justified. For example, when was CTABr used in the synthesis or adsorption? What was the pH of the solution in adsorption? Was the adsorption conducted in a CO2-free atmosphere? Would Pb2+ precipitate as PbCO3 (cerussite) or Pb3(CO3)2(OH)2 (hydrocerussite)? If so, the adsorption capacity would be overestimated. Did the author use XRD to characterize adsorbed solids? Why did not the author conduct an adsorption isotherm to study the adsorption capacity of Pb2+ as we know that the adsorption capacity also depends on adsorbate concentration?
4) All figures and tables were just simply translated into text presentations. I would expect more explanations of why that happens or what possible mechanisms account for such differences or changes. For instance, I wonder why the 518 inverse cm band does not show while a band at 550 inverse cm show in synthesized solids in the FTIR figures and tables. What causes this change?
5) Why are not there any trends of solid properties such as crystallinity, particle size, surface area, and porous volume with increasing synthesis temperature and time? Was the synthesis controlled well?
6) When calculating crystallinity, did the authors use peak height or peak area? Did the authors consider the effects of sample heterogeneity (e.g., packing density) on XRD intensity? Why did not the authors use the refinement method to determine the crystallinity?
7) The quality of all figures is too low, especially the EDX images that are color sensitive.
8) Coma punctuation “,” misuse. The authors used coma as thousand separators in number “343,164,200” in L37 while as decimal points in other numbers of tables. Suggest using period “.” as decimal points.
9) Why did the author only choose sample S80H9 for Pb2+ removal? The synthesis of sodalite was conducted at a variety of temperatures and times, would not one wonder how these solids remove Pb2+?
Other comments
10) Define acronyms such as AAS, SEM, EDX, etc., where they first appear and use their abbreviations thereafter.
11) The authors stated that large particle size results in a decrease in mesoporous volume in L178. Consider plot mesoporous volume vs particle size for a better understanding.
12) There is no Table 4 and Figure 6 while there are two Figure 5’s.
13) Figure 5’s, use the same y-axis scale for a clear comparison between different T’s and with SOD.
14) Check all figure and table references. E.g., Table 1 in L94 should be Table 2, and Figure 4.9 in L212 should be Figure 8.
15) What is the unit of elements in Table 6?
16) The authors may want to talk more about what have been found and what they imply in the Conclusion.
17) Were distilled water and demineralized water the same as deionized water or ultra-pure water?
Author Response
"Please see the attachment."

Reviewer 4 Report
A very routine work. 1. To accurately obtain the coordination number changes of Al and si during synthesis, it is recommended to characterize with 27Al and 29Si NMR. 2. Si/Al of the zeolite should be provided by the ICP methods, 3. The role of the mesoporous for the varied zeolites on the Pb2+ adsorption should be discussed. 4. Fig 8: The element names are not displayed clearly. 5. Since the Al atom was present in the Kaolin (kaolinite, Al4(Si4O10)(OH)8) . Therfore, the The mechanism for exchange of Na+ ions with Pb2+ ions was wrong. nSi-OH + Pb2+ ↔ (Si-O)n-Pb + nH+ (1) The Pb2+ was generally not exchanged with the SiOH. Even if it exists, the chances are very small. 6. The Al amout in the different zeolites should be determined. The relationship between the Al in the zeolite framework and Pb2+ amout should be discussed.Author Response
"Please see the attachment."

Reviewer 5 Report
The manuscript presents the synthesis of a nanoparticles material formed by mesoporous sodalite based on Indonesian kaolin as raw material by using the hydrothermal method through two stages of crystallization using different conditions of temperature and time for the first stage of crystallization. The products of the synthesis were characterized using XRD, FTIR, SEM-EDX and N2 adsorption-desorption. The sodalite synthetized materials were tested as adsorbents for heavy metal Pb 2+ ions removal from wastewater finding that the best adsorption capacity was shown by the material sodalite crystallized for 12 hours at 80 °C
The study was well designed but the presentation of the results needs serious improvements and the language should be carefully revised. In many places the words are not separated by spaces between it.
Examples: page 1, line 24 isnon-biodegradable is non-biodegadable;
line 30: Adsorption with solid adsorbent offers better solution for wastewatertreatment[3]. ….wastewater treatment
Observations:
Some phrases should be reformulated:
Lines 33-34
One of type zeolite, sodalite which consists of 6 rings and 2.8 Å pores size, become a great potential of heavy metal adsorbent.
Lines 41-41
The synthesis parameters including the variation of crystallization time and temperature to control the size and porosity of sodalite.
Equation (1) should be checked and corrected.
Line 89
Which is the catalyst considered?
The denomination of figures 1 and 2 should be revised and the graphic quality of the figures needs to be improved.
BET and BJH abbreviations were not defined in the paper.

Author Response
"Please see the attachment."

Round 2
Reviewer 1 Report
Greetings, Editor thank you for providing me with the opportunity to re-review the article. I reviewed the article with title ``Controlling the size and porosity of sodalite nanoparticles via purification of Indonesian kaolin as efficient adsorbent for heavy metal Pb2+ removal ``. Overall, the article structure and content are now suitable for the MATERIALS journal. I am pleased to send you major level comments, there are some serious flaws which need to be corrected before publication. Generally, I appreciate the author efforts to improve the manuscript. Please consider these suggestions as listed below.
- The title of article is quite long please concise and write in precise way. Please add athletes one introductory statement line in abstract.
- Research gap is still not clear, I got the point, but author should be delivered on more clear way with directed necessity for the beginner. Overall, I agree research gap is fine but write in proper way.
- Introduction section must be written on more quality way, i.e., more up-to-date references addressed. Please target the specific gap such as 2015-2021 etc.
- The novelty of the work must be clearly addressed and discussed, compare previous research with existing research findings and highlight novelty.
- Please add a comparative profile section to compare your results and prove how it better than previous.
- What is the main challenge? Why author choose this material? Please highlight in the introduction part.
- Page 1 Line 31. Please cite this reference with existing reference [1]-Role of nanomaterials in the treatment of wastewater: a review.
- Similar, Page 1 Line 33. Please cite this reference with existing reference [2]- Recent advances in metal decorated nanomaterials and their various biological applications: a review.
- The main objective of the work must be written on the more clear and more concise way at the end of introduction section.
- Please don’t use lumpy reference (such as: [5-7], [8-10], [16-18]). At this point only one reference is enough. Please delete others. Each reference needs to be properly addressed. Please revise your paper accordingly since same issue occurs on several spots in the paper.
- Please check the abbreviations of words throughout the article. All should be consistent. Please revise your paper accordingly since some issue occurs on several spots in the paper.
- Reference 19 should be replaced to this one- Yaqoob, A.A.; Ibrahim, M.N.M.; Ahmad, A.; Reddy, A.V.B. Toxicology and Environmental Application of Carbon Nanocomposite. In Environmental Remediation through Carbon Based Nano Composites; Springer: Berlin/Heidelberg, Germany, 2021; pp. 1–18.
- Reference 20 should be replaced to this one- Umar, K.; Yaqoob, A.A.; Ibrahim, M.; Parveen, T.; Safian, M. Environmental applications of smart polymer composites. Smart Polym. Nanocompos. Biomed. Environ. Appl.2020, 15, 295–320.
- Please add brand specifications, some are missing such as NaOH.
- Regarding the replications, authors confirmed that replications of experiment were carried out. However, these results are not shown in the manuscript, how many replicated were carried out by experiment? Results seem to be related to a unique experiment. Please, clarify whether the results of this document are from a single experiment or from an average resulting from replications. If replicated were carried out, the use of average data is required as well as the standard deviation in the results and figures shown throughout the manuscript. In case of showing only one replicate explain why only one is shown and include the standard deviations.
- Please provide high quality image of figure 1, 8 and 9.
- Section 4 should be renamed by Conclusion and Future perspectives. Conclusion section is missing some perspective related to the future research work, quantify main research findings, highlight relevance of the work with respect to the field aspect. In the present form conclusion is still very weird.
- To avoid grammar and linguistic mistakes, moderate level English language should be thoroughly checked. Please revise your paper accordingly since several language issue occurs on several spots in the paper.
Author Response
"Please see the attachment."

Reviewer 2 Report
Dear Editors,
I do not see detailed answers to my question (point by point). I do not know whether all my questions were answered.
In the abstract, there is no explanation of CTABr….
Author Response
"Please see the attachment."

Reviewer 3 Report
I would expect responses from the authors in a point-by-point fashion, but I do not see that. The authors need to respond to comments by what the responses (answers) are and how they accordingly revise the manuscript. For example, I questioned what the solution pH of the adsorption was, the authors responded by “neutral pH” in the cover letter. The authors should say “The solution pH was around neutral (give a certain value) and added to the manuscript in L## accordingly” and add the pH value in the manuscript. Not only myself wanted to know the solution pH, but also other readers (if it is published) want to see it.
In my previous comment 3, I was concerned that CO2 would cause Pb2+ precipitation in the adsorption experiment. The responses are not convincing, the authors should provide evidence that Pb2+ would not precipitate under the experimental condition, rather than say that the authors did not determine the pH and other studies showed a good adsorption capacity without caring about CO2. At least the authors should provide thermochemical calculations (e.g., Geochemist workbench, Visual Minteq, or Phreeqc) to show no precipitation. To prevent Pb2+ from precipitating as cerussite or hydrocerussite, the authors have to control the pH.
XRD characterization of adsorbed solids is essential to know what has happened in the adsorption. The authors may consider mailing samples to other institutions for XRD analysis.
Again, I suggest doing an adsorption isotherm of Pb2+ on solids, something like the Figure 6 in Iqbal et al.1
Correct grammar errors throughout the manuscript. E.g., add “and” between “non-biodegradable” and “hazardous” in L31, change “have” to “has” in L31, add “and” in front of “reproduction organ” in L 32, add “is” in front of “severely” in L33, and so forth.
Recheck Figure and table references.
Keep color codes consistent in Figure 5a,b for better readability. Why was not T = 50 oC included in Figure 5a? There may not be a clear trend if include T = 50 oC. Is the temperature in Figure 5b 80 oC? Please specify it in the caption. Why was not t = 12 h included in Figure 5b? Figure 5 is misleading, looks like there were good trends of surface area, pore size, and crystallinity as a function of temperature and time, but there were no such trends when including all temperatures and times. Consider reorganizing Figure 5 to include more conditions.
The quality of EDX mapping in Figure 6 is still low. The element labels are not clear.
L1390, there is no Figure 5c,d.
Again, L1558 what is Figure 4.9?
L1562, Table 6 should be Table 5.
Table 5, consider converting weight ratios of elements to molar ratios, which can be compared to sodalite composition. Why did not the authors determine the composition using ICP-OES or ICP-MS? These analyses are more accurate than EDX. For example, the weight % of O is too high. The authors may want to explain why the w.% of O is so high.
Table 6, consider changing “Adsorption capacity of Pb2+ 100 ppm (mg/g)” to “Adsorption capacity (mg/g) of 100 ppm Pb2+”.
L1609, Figure 9 should be Figure 7.
Figure 8, do the authors have data of S80H12 and S50H9? If so, include them in Figure 8.
L1757, “160.8 mg/g”. What is this number? Refer to the table that contains the value.
Table 4 in L1812 and L1821 should be Table 7.
L1813, Figure 8a should be Figure 9a.
L1815-1816, set “2” in “R2” as superscripts.
L1817, provide values of R2 rather than say close to 1.
Table 4 on page 15 (Should be Table 7), correct decimal points. The Sme of SOD60H9 should be 4.29 m2/g. Put the values of SOD in the table for comparison. This table only lists some selected samples for Pb2+ removal, I would expect that the authors could have selected S50H9, S60H9, S70H9, S80H9, and/or S80H3, S80H6, S80H9, and S80H12. Did the authors choose these samples because they had a good increasing trend of surface area? Once again, temperature and time did not affect monotonously the properties of solids (surface area, porous size, particle size, etc.), why?
Now, here is the most concerned question: CTABr was used as a mesopore template to control the particle size and porosity, by which the authors hoped to obtain a better Pb2+ removal (as the title indicates). However, the Pb2+ removal of CTABr treated solids was lower than or similar to that of SOD produced without CTABr (Figure 7, Table 6, Figure 8, and Table 7). The latter solid had a very good Pb2+ removal, what is the point to use CTABr? Controlling the size and porosity of solids using CTABr (actually decreasing surface area and porous volumes) did not improve Pb2+ removal.
References
- Muhammad Iqbal, Asma Saeed, Saeed Iqbal Zafar. FTIR spectrophotometry, kinetics and adsorption isotherms modeling, ion exchange, and EDX analysis for understanding the mechanism of Cd2+ and Pb2+ removal by mango peel waste. Journal of Hazardous Materials 164 (2009) 161–171.
Author Response
"Please see the attachment."

Reviewer 4 Report
The reviewers were completely dissatisfied with the revisions in this version. 1. The exchange mechanism with Pb2+ solely with SiOH was not correct. Because the Al atoms have entered the zeolite framework, most Pb2+ will exchange with the cations bound to the framework. This is the reason why the authors were requested to supplement 27AL NMR and Al elemental analysis. If the author confirms that it only exchanges with SiOH, please provide experimental evidence.
2. Furthermore, the corresponding detailed modifications, which are more modified. Please specify directly in the cover letter.
3. The experiments requested by the reviewers were not completed in time due to covid-19 circumstances. Therefore, whether the article should be rejected or revised and re-submitted up to the editorial discretion.
Author Response
"Please see the attachment."

Reviewer 5 Report
The manuscript was improved. Abstract was reformulated showing better the aim and the main findings of the research. The subchapter 3.2 Adsorption and kinetic study of Pb2+ and Conclusion part were extended and improved.
But, some issues should be signalized.
1)I recommend to present in Introduction section the role of synthesized soladite as catalyst. The study focused on the adsorption capacity of the synthesized soladite or to use Synthesized soladite nanomaterial characterization instead of “Characterization catalyst”.
2)I recommend the revision of the capitation of Figure 1. XRD analysis of samples in variation crystallization time of a) 50-60°C and b) 70-80°C and of Figure 2. FTIR analysis of samples in variation crystallization time of a) 50-60°C and b) 70-80°C
I suggest “XRD analysis of samples with various crystallization times and temperatures a) 50-60°C and b) 70-80 °C”.
3) In Conclusion part the correction of “Na2+ “ in “Na+” must be made.

Author Response
"Please see the attachment."
